# Design of a new multiplex PCR assay for rice pathogenic bacteria detection and its application to infer disease incidence and detect co-infection in rice fields in Burkina Faso

Martine Bangratz[1,2], Issa Wonni[2], Kossi Kini[1,3], Moussa Sondo[1,2], Christophe Brugidou[1,2], Gilles Béna[1], Fatoumata Gnacko[1,2], Mariam Barro[1,2], Ralf Koebnik[1], Drissa Silué[3], Charlotte Tollenaere[1,2]*

1 IRD, Cirad, Univ Montpellier, IPME, Montpellier, France, 2 INERA, Laboratoire de Phytopathologie, LMI PathoBios, Bobo-Dioulasso, Burkina Faso, 3 AfricaRice, Plant Pathology, Bouaké, Ivory Coast

* charlotte.tollenaere@ird.fr

**Data Availability Statement:** All relevant data are within the manuscript and its Supporting Information files.

## Abstract

Crop diseases are responsible for considerable yield losses worldwide and particularly in sub-Saharan Africa. To implement efficient disease control measures, detection of the pathogens and understanding pathogen spatio-temporal dynamics is crucial and requires the use of molecular detection tools, especially to distinguish different pathogens causing more or less similar symptoms. We report here the design a new molecular diagnostic tool able to simultaneously detect five bacterial taxa causing important diseases on rice in Africa: (1) *Pseudomonas fuscovaginae*, (2) *Xanthomonas oryzae*, (3) *Burkholderia glumae* and *Burkholderia gladioli*, (4) *Sphingomonas* and (5) *Pantoea* species. This new detection tool consists of a multiplex PCR, which is cost effective and easily applicable. Validation of the method is presented through its application on a global collection of bacterial strains. Moreover, sensitivity assessment for the detection of all five bacteria is reported to be at 0.5 ng DNA by μl. As a proof of concept, we applied the new molecular detection method to a set of 256 rice leaves collected from 16 fields in two irrigated areas in western Burkina Faso. Our results show high levels of *Sphingomonas* spp. (up to 100% of tested samples in one field), with significant variation in the incidence between the two sampled sites. *Xanthomonas oryzae* incidence levels were mostly congruent with bacterial leaf streak (BLS) and bacterial leaf blight (BLB) symptom observations in the field. Low levels of *Pantoea* spp. were found while none of the 256 analysed samples was positive for *Burkholderia* or *Pseudomonas fuscovaginae*. Finally, many samples (up to 37.5% in one studied field) were positive for more than one bacterium (co-infection). Documenting co-infection levels are important because of their drastic consequences on epidemiology, evolution of pathogen populations and yield losses. The newly designed multiplex PCR for multiple bacterial pathogens of rice is a significant improvement for disease monitoring in the field, thus contributing to efficient disease control and food safety.

**Funding:** This work was publicly funded through ANR (the French National Research Agency) under «Investissements d'avenir» programme with the reference ANR-10-LABX-001-01 Labex Agro (E-Space and RiPaBIOME projects), coordinated by Agropolis Fondation under the frame of I-SITE MUSE (ANR-16-IDEX-006) and by THE CGIAR Research Program on Rice Agri-food Systems (RICE). The funders had no role in study design, data collection and analysis, decision to publish, or preparation of the manuscript.

**Competing interests:** The authors have declared that no competing interests exist.

# Introduction

Over the last 20 years, West Africa has specifically experienced a large surge in rice consumption, with average increase of 4.6% each year [1]. This is a consequence of demographic growth, but also of habit changes resulting from urbanization, where the people prefer fast-prepared food such as rice, compared to other cereals such as maize, millet or sorghum. With such a growing demand, local rice production, which accounted for 80% of the demand in the 1960s, has now dropped to only 60% [1]. In reaction to the 2008 food crisis, West African states are developing ambitious projects to increase local rice production and to decrease their dependency to worldwide rice market [2]. Areas cultivated with rice have increased dramatically and rice cultivation is intensifying [3, 4]. In Burkina Faso, rice-growing areas increased by three-fold between 2006 and 2010 (FAOSTAT database), thanks to the additional areas developed to grow rainfed lowland rice.

The increase of rice-growing areas and intensification of rice production constitute major agricultural changes, and such modifications can make rice more vulnerable to several diseases [5]. According to Savary et al. 2019, global yield losses due to pathogens and pests in rice are estimated at 30% (range: 24.6–40.9%) worldwide and were even higher in sub-Saharan Africa [6]. In particular, several diseases caused by bacterial pathogens can significantly reduce rice yields.

Major bacterial diseases of rice are Bacterial Leaf Blight (BLB) and Bacterial Leaf Streak (BLS), caused by two pathovars of *Xanthomonas oryzae*: *X. oryzae* pv. *oryzae* (*Xoo*) and *X. oryzae* pv. *oryzicola* (*Xoc*), respectively. Yield loss due to BLB may reach 50% in susceptible varieties under favorable environmental conditions [7]. In West Africa, BLB was first reported in Mali in 1979 [8]. BLS was first detected in Mali in 2003 [9] and 2009 in Burkina Faso [10] and is now considered as an emerging disease [11]. Promising sources of resistance to these bacterial diseases were identified [12] but the deployment of efficient control strategies based on genetic resistances relies on profound knowledge of the epidemiological situation in the targeted locations.

In addition to *X. oryzae*, different species of *Pantoea* [13], and *Sphingomonas* [14] genera were recently described to be responsible for BLB-like symptoms in rice. Either *Pantoea stewartii*, *P. ananatis* and P. *agglomerans* were identified as causing leaf blight disease in different countries worldwide [15, 16] and in particular in Africa [17, 18]. The taxonomy of *Sphingomonas* species causing BLB-like symptoms have also been detected on rice seeds from eight African countries [14], but only a few isolates were described as plant pathogens. *Sphingomonas* and *Pantoea spp*. have been frequently isolated from rice seeds in India [19] and Africa [20]. The importance of *Pantoea* and *Sphingomonas* species as rice pathogens in West Africa, compared to the well-known devastating *Xanthomonas* bacteria, remains to be documented and specific molecular detection tools are critical to this purpose.

The species *Pseudomonas fuscovaginae* is responsible for bacterial sheath brown rot of rice. However, other pathogens, such as *Sarocladium oryzae* and *Fusarium* spp., cause similar sheath rot symptoms [21]. Bacterial sheath brown rot due to *P. fuscovaginae* is seed-transmitted. It has so far not been reported in West Africa (CABI 2007, cited by [21]) and is generally associated with high elevation areas (Madagascar [22], East and Central Africa), but it was later found as well in lowlands [21–23].

*Burkholderia glumae* causes bacterial panicle blight of rice, which is an increasingly important disease problem in global rice production [24], especially in the context of global warming and change in environmental conditions [25]. *B. gladioli* causes a very similar disease of the panicle [26], both species reducing root development, grain weight and inducing inflorescence sterility [27]. Presence of *B. glumae* on rice in Africa has been reported twice (Burkina Faso

[28] and in South Africa [29]), but with no molecular data permitting to confirm the taxonomic affiliation. The high occurrence of *B. glumae* in Asia and Latin America, however, make the development of an efficient diagnostic tool to detect the disease urgent if it were to gain importance in Africa.

Plant disease detection methods are essential for epidemiological surveillance and to facilitate effective management practices [30]. In rice, bacterial pathogens are mostly detected using specific molecular methods focusing on a single genus. In particular, efficient detection protocols are available for *X. oryzae* [31] and for *Pantoea* spp. [32]. Few recently introduced methods can simultaneously detect several rice pathogens. One targets three species/pathovars, *Xoo*, *Xoc* and *B. glumae* [33]. Another one focused on three bacterial seed-borne diseases caused by *B. glumae*, *Xoo* and *Acidovorax avenae* subsp. *avenae* [34]. A third one was designed for simultaneous detection of six bacterial pathogen of rice: *Xoo*, *Xoc*, *P. fuscovaginae*, *B. glumae*, *B. gladioli* and *A. avenae* subsp. *avenae* [35]. None of existing multiplex PCR assay have focused on the simultaneous detection of these pathogens in Africa. Also none of them included the two bacterial genera *Pantoea* and *Sphingomonas* that were recently shown to cause severe symptoms on rice, in West Africa [14, 17, 18], but also in other countries [36–38].

We report here the design a simple multiplex PCR scheme, allowing the simultaneous detection of five bacterial taxa causing important diseases on rice in Africa: (1) *Pseudomonas fuscovaginae*, (2) *Xanthomonas oryzae*, (3) *Burkholderia glumae* and *Burkholderia gladioli*, (4) *Sphingomonas* and (5) *Pantoea* species. We validate the new method on a global collection of bacterial strains and we show an example of application with the estimation of incidence levels of three bacterial taxa in two irrigated areas in western Burkina Faso.

## Material and methods

### Development of new specific DNA primers for *Pantoea*, *Sphingomonas* and *Burkholderia spp*

Table 1 presents the set of primers used that were designed for the diagnostic test. We used various primers from the literature for the detection of *X. oryzae* [31] and *P. fuscovaginae* [35]. For other targeted pathogens, we designed new primers in a way that the different DNA fragments could be easily separated and identified by conventional agarose gel electrophoresis following their multiplex PCR amplification.

For the design of *Pantoea* spp. specific primers, we targeted a region in the ATP synthase subunit beta AtpD (*atpD*) gene, which had been exploited for a diagnostic multiplex PCR scheme for three species of plant-pathogenic *Pantoea* species [32]. Similarly, *Sphingomonas*-

**Table 1. List of the primers used for the multiplex PCR for the detection of multiple bacterial diseases in rice.**

| Targeted pathogen | Primer's name | Primer's sequences (5′–3′) | Fragment size (bp) | Reference |
|---|---|---|---|---|
| *Pseudomonas fuscovaginae* | Pfs207-F | CAGTTCGATGGTCTGGGAAT | 710 | Cui et al., 2016 [35] |
| | Pfs207-R | GGGACTGGTAAAGCACGGTA | | |
| *Burkholderia glumae* and *B. gladioli* | toxB_F | GCATTTGAAACCGAGATGGT | 508 | G. Béna, this study |
| | toxB_Rd | TCGCATGCAGATAACCRAAG | | |
| *Sphingomonas* spp. | Sphingo_KK_F1 | CGGCTGCTAATACCGGATGAT | 435 | K. Kini, this study |
| | Sphingo_KK_R1 | AGGCAGTTCTGGAGTTGAGC | | |
| *Xanthomonas oryzae* | Xo3756F | CATCGTTAGGACTGCCAGAAG | 331 | Lang et al., 2010 [31] |
| | Xo3756R | GTGAGAACCACCGCCATCT | | |
| *Pantoea* spp. | PAN_KK263F | GCGAGCCAATCGACATTA | 263 | K. Kini, this study |
| | PAN_KK263R | CGAGTAACCTGAGTGTTCAG | | |

specific primers were designed for the 16S rRNA gene, which had been used for diagnosis of several *Sphingomonas* species on symptomatic rice leaf samples from eight West African countries [14].

Specific primers for both *Burkholderia glumae* and *B. gladioli* were designed within the toxoflavin biosynthesis operon. Toxoflavin is a phytotoxin and has been so far detected only in the two closely related species *B. glumae* and *B. gladioli* (Béna, pers. com.). It is encoded by a cluster of eleven genes, for both production and secretion of the toxin. We took advantage on the very low nucleotide diversity among all the genomes available so far to design primers within *ToxB* that is predicted to encode a GTP cyclohydrolase II. We selected one pair of primers with a nearly 100% identity among all sequences available in GenBank (one degenerated site in the reverse primer) resulting in the amplification of a 508 bp fragment.

## Multiplex PCR assay optimization

Following the optimization, the final conditions for the multiplex PCR protocol were: 2 µl of DNA added to 23 µl master mix comprising 5 µl Hot Firepol Multiplex Mix ready to load 5X (Solis BioDyne,Tartu, Estonia), 1.25 µl $(NH_4)_2SO_4$ (160 mM), 0.2 µl of each primers specific of *P. fuscovaginae* at 5 µM, 0.2 µl of each primers specific of *B. glumae* and *B. gladioli* at 100 µM, 0.3 µl of each primers specific of *Pantoea* spp. at 100 µM and 0.3 µl of each primers specific of *X. oryzae* at 10 µM and finally 0.1 µl of each primers specific of *Sphingomonas* spp. at 10 µM.

DNA amplification was performed with an Applied Biosystems Veriti 96-Well Thermal Cycler and the following cycles were: 12 min activation at 95˚C, 30 cycles of 94˚C for 30 sec, 58˚C for 30 sec and 72˚C for 45 sec and a final extension of 7 min at 72˚C. Aliquots (10 µl) of PCR-amplified DNA were analyzed by agarose gel electrophoresis at 100 V for 90 minutes in 0.5X TBE buffer. The size of amplified PCR products was determined by comparing to a 100 bp DNA ladder (Solis BioDyne, Tartu, Estonia). Positive (1 ng DNA of each bacteria) and negative (water) controls were included in each reaction. Multiplex PCR protocol is available at dx.doi.org/10.17504/protocols.io.bcpaivie.

## Validation of the multiplex PCR assay by screening a bacterial collection

We applied the newly designed detection test for a collection of 42 isolates of bacteria from 12 different countries. The number of the tested bacterial strains and the countries from which they originated are shown in Table 2. The exhaustive list of isolates is given in S1 Table.

**Table 2. Bacterial strains of each targeted taxon used to validate the detection protocol.** For each taxon, the strain used as reference strain for most validation tests appears in bold.

| Targeted pathogen | Species / pathovar | Strains per country | Total number of strains tested |
|---|---|---|---|
| *Pseudomonas fuscovaginae* | | Mexico (UPB0526); Philippines (**UPB0735**); Madagascar (UPB0736); Colombia (UPB0896) | **4** |
| *Sphingomonas* spp. | | Benin (ASP6, ASP26); Burkina Faso (**V1-2**, ASP109); Mali (ASP111, ASP116, ASP128); Nigeria (ASP621, ASP641); Togo (ASP160, ASP204) | **11** |
| *Xanthomonas oryzae* | *X. oryzae* pv. *oryzae* | Burkina Faso (BAI3); Mali (ABB27, ABB37, ABB42, ABB43, ABB44); Philippines (PXO99) | **10** |
| | *X. oryzae* pv. *oryzicola* | Burkina Faso (**BAI10**, BAI119); Philippines (BLS256) | |
| *Pantoea* spp. | *P. ananatis* | Benin (ARC22); Burkina Faso (ARC315); Burundi (ARC593) | **9** |
| | *P. stewartii* | Benin (ARC903); Nigeria (**ARC10**) | |
| | *P. agglomerans* | Benin (ARC982, ARC1000); Nigeria (ARC282); Togo (ARC933) | |
| *Burkholderia* spp. | *B. glumae* | Viet-Nam (**NCPPB3923**); Japan (LMG2196, CFBP3831); Colombia (3252–8) | **8** |
| | *B. gladioli* | Colombia (ABIP15, ABIP49, ABIP128, ABIP 173) | |

Bacterial isolates were grown on Glucose Yeast Peptone medium for 48 h. DNA extraction was performed on the obtained colonies using Wizard® Genomic DNA Purification Kit (Promega, Madison, Wisconsin, USA), following the manufacturer's recommendations. Multiplex PCR was performed following the protocol described above. All experiments were repeated twice.

### Applying multiplex PCR assay to a set of rice leaves samples from Burkina Faso

The newly designed detection test was applied to a collection of leaves sampled in 2016 in two irrigated rice-growing areas in western Burkina Faso: Banzon (GPS coordinates: N 11.31955; 04.80978) and Karfiguela (GPS coordinates: N 10.68347; W 04.81605). This sample collection follows up the prospections performed in 2015 and described in [39]. We visited all the fields between October 10 and November 30, when rice plants were at maximum number tillage or panicle initiation stage. A regular sampling strategy was adopted with the sampling of 16 plants per field, following a 4x4 grid within 20x20 meters fields. For each of the 16 plants (labelled from A1 to D4), we sampled three leaves (including symptomatic leaves if observed) and kept them dry using a plastic bag containing silica gel for subsequent molecular analyses. Symptom-based incidence was estimated in each field for BLB and BLS by carefully observing plants in the four cells along the diagonal of the 4x4 grid (obtained average incidence resulted from the average of recorded incidence levels over the four cells). Eight fields were studied at each site (16 fields in total over the two sites), the total number of sampled plants consequently being 256 (16 plants in each of eight studied fields at two sites). In every case, we obtained permission from the farmers to work and sample leaves in their fields.

DNA extraction was performed for each sample with approximately 20 mg of dried leaves. Samples were ground using the Tissue Lyser II (Qiagen, Inc., Valencia, CA) until a fine powder was obtained. DNA extraction was carried out as described by Li et al. 2008 [40] except that the 2-mercaptoethanol was substituted by 0.5% sodium bisulfite and that the samples were put at -20° C during 30 min after adding isopropanol. We dissolved the pellet in 50 μL sterile water. Obtained DNA was diluted 1:2 and each reaction was performed with 2 μL of DNA. Nucleic acids extraction protocol is available at dx.doi.org/10.17504/protocols.io.bcntiven.

Data were analysed using R software [41] and a map reporting diseases incidences was built using Qgis [42]. We used generalised linear mixed model (GLMM) with the library lme4 to test for an effect of the site (two distinct irrigated areas) on the presence/absence of *Sphingomonas* spp., including the considered field as random factor.

## Results

### Optimization of the molecular diagnostic protocol

The multiplex PCR protocol allowing for the simultaneous detection of *P. fuscovaginae*, *X. oryzae*, *Burkholderia* (both *B. glumae* and *B. gladioli*) as well as *Sphingomonas* and *Pantoea* spp. (Fig 1) was used.

We first tested the specificity and the sensitivity of the molecular diagnostic protocol for the simultaneous detection of *P. fuscovaginae*, *X. oryzae*, *Burkholderia* (both *B. glumae* and *B. gladioli*) as well as *Sphingomonas* and *Pantoea* spp. (Fig 1). The concentration of $(NH_4)_2SO_4$ (S1 Fig), number of PCR cycles and the annealing temperature were optimized to avoid nonspecific DNA amplification. Addition of 250 ng plant DNA to bacterial DNA did not change the amplification results.

The sensitivity of the multiplex PCR was slightly lower than in simplex PCR under the same conditions (Fig 2). Multiplex PCR was able to detect *B. glumae* and *B. gladioli* at 50 pg/μl,

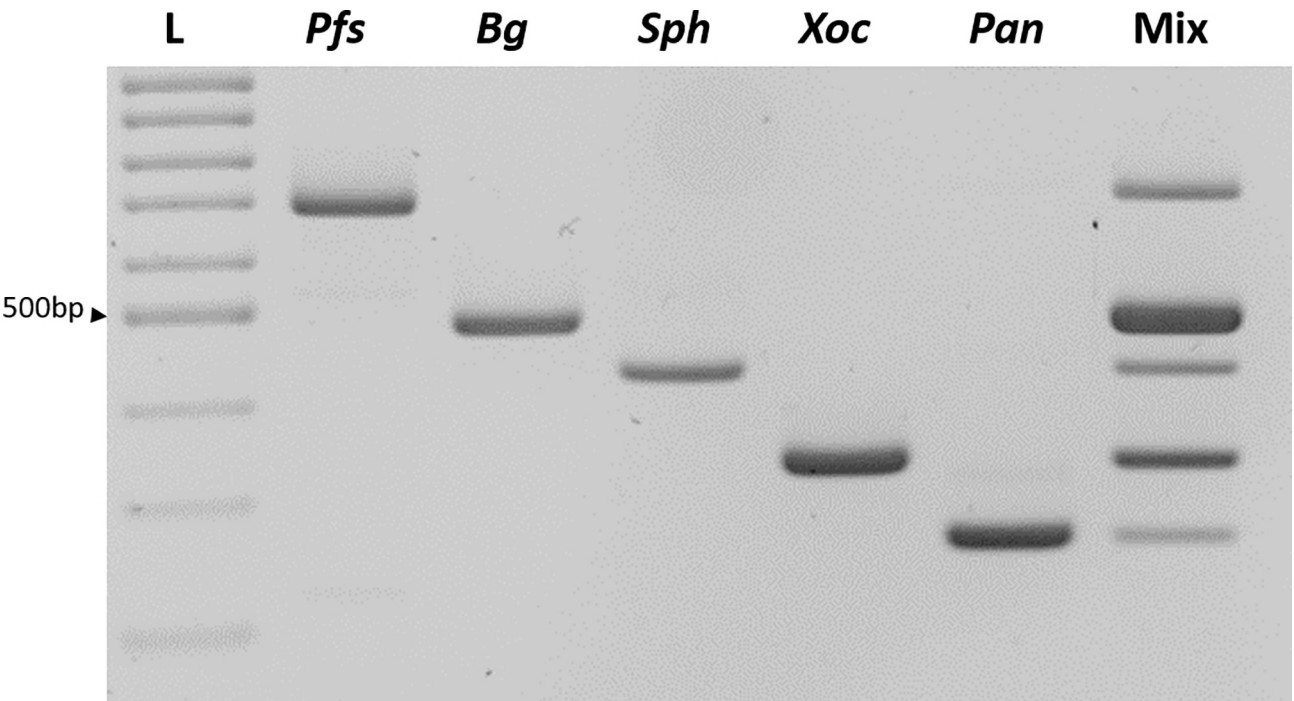

**Fig 1. Detection of the five bacterial taxa using the newly described multiplex PCR protocol.** L: molecular size marker, 100 bp DNA ladder ready to load, Solis Biodyne, *Pfs*: *P. fuscovaginae* strain UBP735, *Bg*: *B. glumae* strain NCPPB 3923, *Sph*: *Sphingomonas* strain V1-2, *Xoc*: *X. oryzae* pv. *oryzicola* strain BAI10, *Pan*: *Pantoea* strain ARC10, Mix: Equal amounts of all five DNA samples.

while 0.5 ng/µl was required for the other four bacteria (*P. fuscovaginae*, *Sphingomonas* spp., *X. oryzae* and *Pantoea* spp., Fig 2).

## Application of the molecular diagnostic on dried leaves collected in the field in western Burkina Faso

Nucleic acid extraction was performed on the 256 samples. The average concentration obtained was 271 ng/µL (minimum 33 ng/µL, maximum 932 ng/µL). Multiplex PCR was

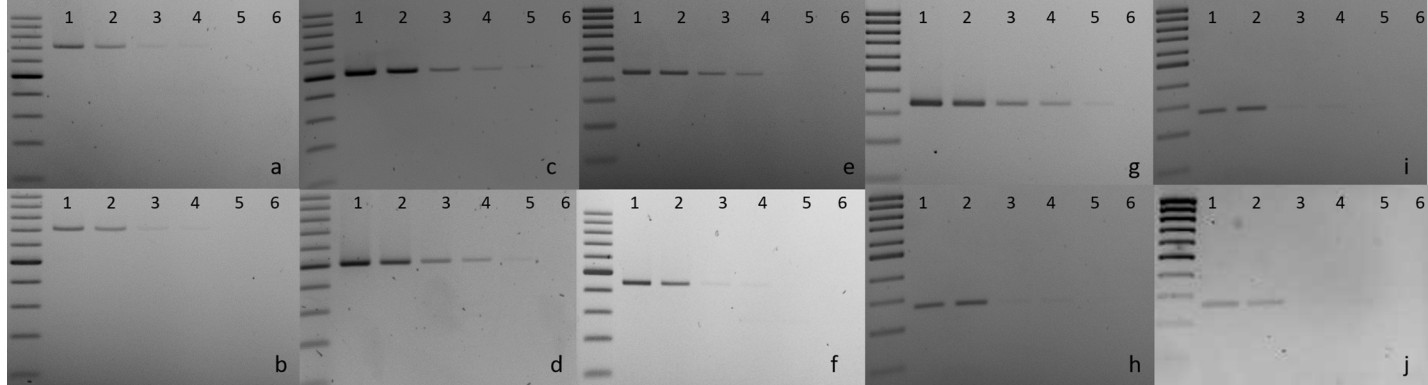

**Fig 2. Sensitivity of the newly described multiplex PCR method compared to the corresponding simplex PCR for each of the targeted bacterial taxon.** Every reaction was performed with six samples of corresponding control bacteria at different concentrations. Lane 1: 1 ng/µl, lane 2: 0.5 ng/µl, lane 3: 0.1 ng/µl, lane 4: 0.05 ng/µl and lane 5: 0.01ng/µl, lane 6: water control. a-b: *P. fuscovaginae* strain UBP735, c-d: *B. glumae* strain NCPPB 3923, e-f: *Sphingomonas* strain V1-2, g-h: *X. oryzae* pv. *oryzae* strain BAI10, i-j: *Pantoea* strain ARC10. a, c, e, g, i: simplex PCR with only one primer pair in each case. b, d, f, h, j: multiplex PCR including the five primer pairs.

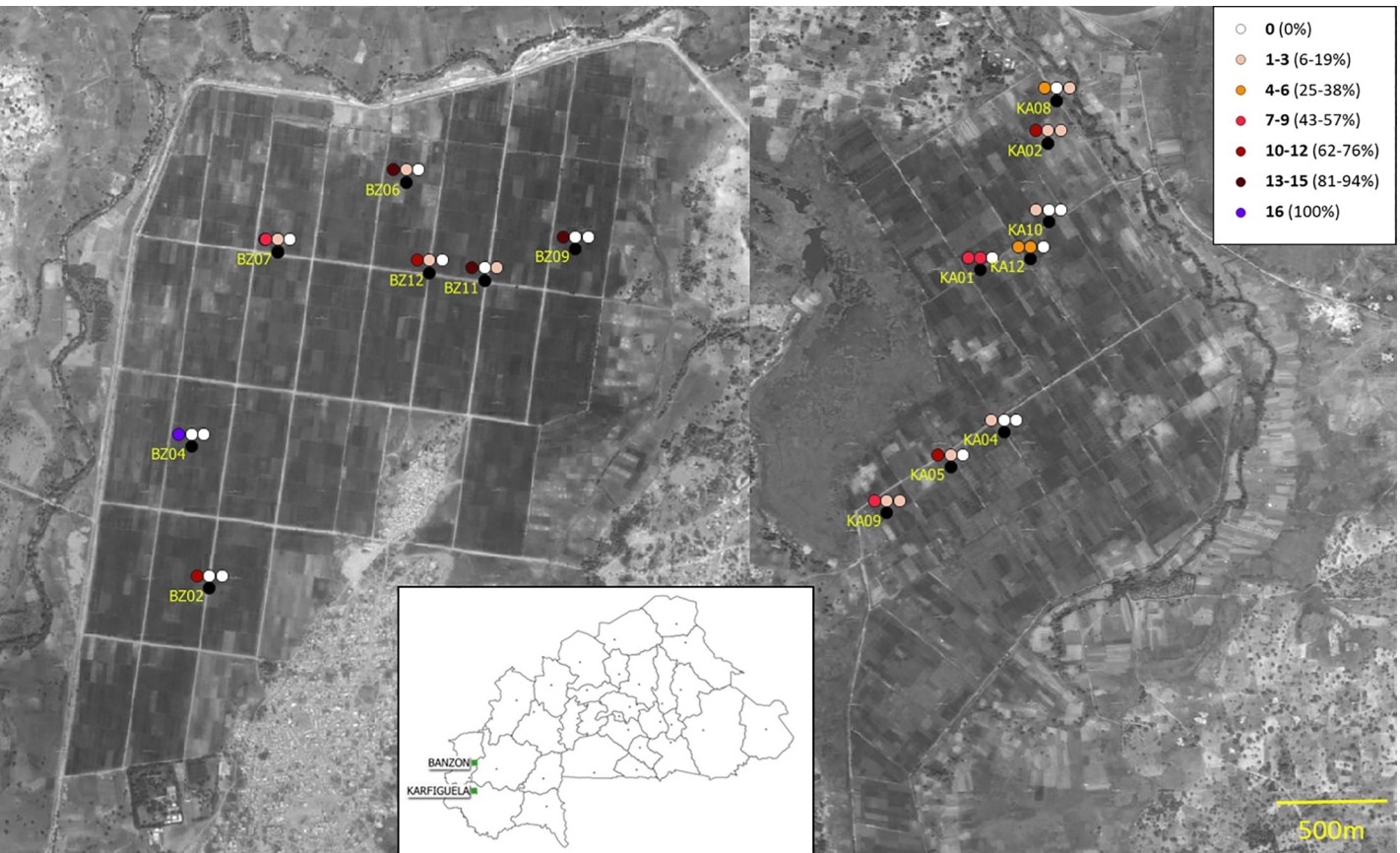

**Fig 3. Incidence of targeted bacterial taxa in two irrigated areas located in western Burkina Faso.** On the left the irrigated area of Banzon and on the right the irrigated area of Karfiguela. For each studied field, the black dot correspond to the location of the field; incidences for *Sphingomonas* spp., *X. oryzae*, and *Pantoea* spp. are indicated by colored dots from left to right. Gradient of red color indicates increased frequency of positive samples (incidence).

performed twice on all samples and the repeatability of the results was 91.5% (140 identical results over 153 positive results). We preferred to adopt a conservative approach and considered all the 13 ambiguous results as negative.

*P. fuscovaginae*, *B. glumae* and *B. gladioli* were never detected in any of the 256 samples. On the other hand, *Sphingomonas* spp., *X. oryzae* and *Pantoea* spp. were found in 153 (59.8%), 22 (8.6%), and 5 (2.0%) of the 256 samples, respectively. Gel pictures of a few samples as examples appear in S2 Fig. Detailed incidences of the three pathogens are presented in Fig 3, Table 3 and S3 Fig. The average *Sphingomonas* spp. disease incidence varied between the two studied sites ("Site" effect, chi = 11.2, $p < 0.001$) with 78.9% of positive plants in Banzon compared to 40.6% in Karfiguela.

We also assessed multiple infection levels and found that at least two of the three detected bacterial taxa were found in 23 (9.0%) out of the 256 samples, with *Sphingomonas* always one of the taxa detected. Two plants (0.8%) simultaneously presented all three taxa. Among the 153 samples detected as positive for *Sphingomonas* spp., 23 (15.0%) were also found positive for at least one other bacterium.

Among the 256 plant samples, seven harbored BLS symptoms and five of them were positive for *X. oryzae*. At the field level, the two fields (KA01 and KA12) that had the highest BLS incidence estimated through symptom observation (95.3% and 40.0%) were also the fields with highest frequency of *X. oryzae*-positive samples among the 16 analyzed plants (43.8% and

**Table 3. Obtained results in the 16 fields surveyed in Southern Burkina Faso: Pathogen incidences derived from the use of the developed molecular diagnostic tool on 16 sampled plants per field, and disease incidence estimated from symptom observations in four cells of the field's diagonal.**

| Studied site | Molecular diagnostic results: Number of positive samples (percentage of studied plants) | | | | Estimated disease incidence based on symptom observations | |
|---|---|---|---|---|---|---|
| | *Sphingomonas* spp. | *Xanthomonas oryzae* | *Pantoea* spp. | Multiple infection of at least 2 out of the 3 bacteria | BLB | BLS |
| BZ02 | 11 (68.8%) | 0 | 0 | 0 | 15% | 0 |
| BZ04 | 16 (100.0%) | 0 | 0 | 0 | 1% | 0 |
| BZ06 | 14 (87.5%) | 3 (18.8%) | 0 | 3 (18.8%) | 10% | 1% |
| BZ07 | 14 (87.5%) | 0 | 0 | 0 | 1% | 0 |
| BZ09 | 13 (81.3%) | 0 | 0 | 0 | 15% | 0 |
| BZ10 | 8 (50.0%) | 1 (6.3%) | 0 | 1 (6.3%) | 7% | 0 |
| BZ11 | 13 (81.3%) | 0 | 2 (12.5%) | 2 (12.5%) | 6% | 0 |
| BZ12 | 12 (75%) | 3 (18.8%) | 0 | 3 (18.8%) | 11% | 0 |
| **BANZON** | **101 (78.9%)** | **7 (5.5%)** | **2 (1.6%)** | **9 (7.0%)** | **-** | **-** |
| KA01 | 9 (56.3%) | 7 (43.8%) | 0 | 6 (37.5%) | 0 | 95% |
| KA02 | 11 (68.8%) | 1 (6.3%) | 1 (6.3%) | 1 (6.3%) | 0 | 0 |
| KA04 | 2 (12.5%) | 0 | 0 | 0 | 0 | 0 |
| KA05 | 12 (75.0%) | 2 (12.5%) | 0 | 2 (12.5%) | 1% | 0 |
| KA08 | 4 (25.0%) | 0 | 1 (6.3%) | 1 (6.3%) | 0 | 0 |
| KA09 | 7 (43.8%) | 1 (6.3%) | 1 (6.3%) | 1 (6.3%) | 17% | 0 |
| KA10 | 3 (18.8%) | 0 | 0 | 0 | 0 | 0 |
| KA12 | 4 (25.0%) | 4 (25.0%) | 0 | 3 (18.8%) | 1% | 40% |
| **KARFIGUELA** | **52 (40.6%)** | **15 (11.7%)** | **3 (2.3%)** | **14 (10.9%)** | **-** | **-** |
| **TOTAL** | **153 (59.8%)** | **22 (8.6%)** | **5 (2.0%)** | **23 (9.0%)** | **-** | **-** |

BLB: Bacterial Leaf Blight symptoms; BLS: Bacterial Leaf Streak

25.0%, respectively, Table 3 and S4 Fig). Apart from these two fields, the highest *Xo* incidence levels (3/16 = 19%) were found in two fields (BZ06 and BZ12) also has relatively high symptom-based BLB estimates (ca 10%; Table 3 and S4 Fig). We found no clear relationship between BLB symptom-based incidence estimates and either *Sphingomonas* or *Pantoea* molecular incidence estimates (Table 3).

## Discussion

In the present study, we designed an efficient multiplex PCR method allowing the rapid and simultaneous detection of several important bacterial rice pathogens. The method worked accurately either using bacterial cultures or rice leaves collected in the field. This new molecular detection test was validated on 42 strains of five bacterial taxa (Table 2). The simple and cost-effective CTAB-based protocol enabled extraction of good-quality DNA from rice leaves that was well-suited for subsequent PCR amplification. The designed multiplex PCR required only one reagent: a commercial MasterMix that is cheap and stable at ambient temperature. All this makes this protocol easy-to-use in different labs including those located in low-income countries. With a focus on the major bacterial rice diseases found in Africa, in particular the recently described *Pantoea* and *Sphingomonas* spp., this tool will complement already available detection protocols [31–35] for improved diagnostics of bacterial rice diseases and epidemiological surveillance in the field.

As a proof of concept, we applied the new assay on a set of 256 samples of rice leaves collected in Burkina Faso rice fields in 2016. Neither *Burkholderia glumae* and *gladioli* nor *Pseudomonas fuscovaginae* were detected in any analyzed samples. The test being highly sensitive, we are confident that these two bacterial taxa were absent from our samples. The sampling protocol (three leaves collected at maximum tiller number / flowering initiation stage) may partly be responsible for these absences of sheat and grain-associated bacteria, although *P. fuscovaginae* was shown to behave as endophytic bacteria [43]. To date, *P. fuscovaginae* has not been reported in West Africa (CABI 2007, cited by [21]) and reports of *B. glumae* in Africa have not been confirmed through molecular data. Detecting any of these two taxa in our samples would consequently have been a surprise, but it was important to include them in the multiplex PCR scheme for further applications on any rice tissue.

We found *Sphingomonas* spp. to be highly frequent, with almost 60% of the rice leaves being positive. However, contrasting average incidence results were obtained in the two studied sites for this pathogen, with higher incidence in the irrigated area of Banzon (78.9%) than that of Karfiguela (40.6%). Further work is required to infer the drivers of this spatial heterogeneity. In addition, the genus *Sphingomonas* and its pathogenicity remain poorly investigated. Preliminary 16S sequencing analyses of eight African strains showed their membership to the genus *Sphingomonas*, as they belong to a phylogenetic group that includes many type strains including *S. paucimobilis*, *S. melonis* and *S. zeae* [14]. Further work is required to describe *Sphingomonas* phylogeny and identify pathogenic species on rice and other crops.

Analysis of microbial content of 1916 rice seed samples originating from eleven African countries and received at the Plant Quarantine Unit of AfricaRice Research Station between 2013 and 2016 [20] revealed a high incidence of *Pantoea* and *Sphingomonas* (in Benin for example, average 27% and 18.5% infected seeds respectively were detected). Although seed pathogen loads were not tested in this study, we are confident that the newly developed tool will be useful for evaluating the sanitary status of rice seeds.

Molecular-based incidence estimates of *X. oryzae* are globally congruent with symptom observations. Indeed, the two fields presenting the highest molecular-based incidence (KA01 and KA12) were also the ones where we found highest symptom-based incidence for BLS (Table 3 and S4 Fig). BLS symptoms, caused by *Xoc*, are more specific and easy-to-diagnose than those of BLB, which can be due to *Xoo*, but also due to *Sphingomonas* or *Pantoea* spp. Our results suggest that part of the BLB symptoms may be due to *Xoo* in several fields (Table 3 and S4 Fig), but additional work is required to decipher relative importance of the other bacterial taxa causing BLB in Africa. Here, the newly designed detection test will certainly be extremely helpful.

Although only three out of the five targeted bacterial taxa were detected in analyzed field samples, levels of co-infection were found to be relatively high. Indeed, the samples that were positive for at least two bacteria represented 8% of the total dataset, and up to 37.5% in one particular studied field. Co-infection can significantly affect symptom expression and within-plant pathogen multiplication, as well as epidemiology and the evolution of pathogen populations [44, 45]. Epidemiological survey of bacterial diseases in rice in Burkina Faso will allow a better understanding of the epidemiology of each disease taken separately but also to decipher the interactions between pathogens and their consequences on yield losses and epidemiology.

With the ongoing global climate and trade changes, risks of crop disease outbreaks increase and make it even more crucial to have efficient disease detection protocols for the major diseases of important staple crops such as rice. Targeting multiple pathogens at once is a fast and cost-effective strategy to get an exhaustive view on infection status of a sample. Besides this multi-bacteria detection tool, perspectives include the addition of viral and/or fungal primers (into the same or a second multiplex PCR scheme) in order to get a more global information.

Such diagnostic tools are key components of a global epidemiological surveillance system for crop diseases [46], which is required to implement effective control strategies contributing to food safety.

## Supporting information

**S1 Table. Characteristics of all tested bacterial isolates.**
(XLSX)

**S2 Table. Raw data for the application of the detection test to a set of 256 rice leaves samples collected in western Burkina Faso in 2016.**
(XLSX)

**S1 Fig. Effect of the addition of $(NH_4)_2SO_4$ the on the sensitivity of the multiplex PCR method.** a: multiplex PCR with $(NH_4)_2SO_4$; b: multiplex PCR without $(NH_4)_2SO_4$. Every reaction was performed with a mix of six samples of each control bacterial strain (*Pseudomonas fuscovaginae* strain UBP735, *Burkholderia glumae* strain NCPPB 3923, *Sphingomonas* spp. strain V1-2, *Xanthomonas oryzae* pv. *oryzae* strain BAI10, *Pantoea* ssp. strain ARC10) at different concentrations. Lane 1: 5 ng/µl, lane 2: 1 ng/µl, lane 3: 0.5 ng/µl, lane 4: 0.1 ng/µl and lane 5: 0.05ng/µl, lane 6: water control.
(TIF)

**S2 Fig. Results of the detection method for a few field-collected leaves samples.** Example of detection of different bacterial taxa from field samples. Six samples were chosen to present the different possibilities obtained. L: molecular size marker, 100pb DNA ladder ready to load, Solis Biodyne. M: all five bacterial DNA samples.
(TIF)

**S3 Fig. Incidence of targeted bacterial taxa in two irrigated areas located in western Burkina Faso.** Each panel corresponds to one of the targeted bacterial taxa. For each of the two studied irrigated areas (Sites: Banzon and Karfiguela), both the boxplot, as well as the points corresponding to incidence estimates for each field, are given.
(TIF)

**S4 Fig. Relationship between *Xanthomonas oryzae* (*Xo*) incidences derived from the use of the developed molecular diagnostic tool on 16 sampled plants per field, and Bacterial Leaf Blight (BLB) and Bacterial Leaf Streak (BLS) disease incidence estimated from symptom observations in four cells of the field's diagonal.** Each point corresponds to one field, with BLS incidence estimate based on symptom observations on the x-axis and *Xo* incidence based on molecular detection on the y-axis (the red dotted line representing the linear regression between the two variables). Color of the points reflects BLB incidence estimate based on symptom observations.
(TIF)

**S1 Raw images.**
(PDF)

## Acknowledgments

We are very grateful to Sylvain Zougrana, Abalo Itolou Kassankogno, Nils Poulicard and Martial Kabore for their contribution to the sampling in Burkina Faso in 2016. We thank the rice farmer's from Banzon and Karfiguela for their collaboration.

We thank Philippe Petit for conducting some of the molecular biology analyses and Boris Szurek for constructive discussions overall the project. We are grateful to Johanna Echeverri and the Experimental Center Las Lagunas-Fedearroz in Colombia for providing *Burkholderia gladioli* isolates and to Claude Bragard for providing the *Pseudomonas fuscovaginae* strains and for helpful comments on the manuscript.

Part of this work was performed thanks to the facilities of the "International joint Laboratory PathoBios: Observatory of plant pathogens in West Africa: biodiversity and biosafety" (www.pathobios.com; @PathoBios).

## Author Contributions

**Conceptualization:** Martine Bangratz, Issa Wonni, Kossi Kini, Ralf Koebnik, Drissa Silué, Charlotte Tollenaere.

**Data curation:** Martine Bangratz, Moussa Sondo, Mariam Barro.

**Formal analysis:** Charlotte Tollenaere.

**Funding acquisition:** Christophe Brugidou, Charlotte Tollenaere.

**Investigation:** Martine Bangratz, Kossi Kini, Moussa Sondo, Gilles Béna, Fatoumata Gnacko, Mariam Barro.

**Methodology:** Martine Bangratz, Issa Wonni, Kossi Kini, Gilles Béna, Drissa Silué.

**Project administration:** Issa Wonni, Christophe Brugidou, Charlotte Tollenaere.

**Supervision:** Martine Bangratz, Issa Wonni, Drissa Silué, Charlotte Tollenaere.

**Writing – original draft:** Charlotte Tollenaere.

**Writing – review & editing:** Martine Bangratz, Kossi Kini, Christophe Brugidou, Gilles Béna, Ralf Koebnik, Drissa Silué.

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
