## [Decision Letter · Decision Letter 0]

28 Jan 2020

PONE-D-19-32664

Design of a new multiplex PCR assay for rice pathogenic bacteria detection and its application to infer disease incidence and detect co-infection in rice fields in Burkina Faso

PLOS ONE

Dear Dr Tollenaere,

Thank you for submitting your manuscript to PLOS ONE. After careful consideration, we feel that it has merit but does not fully meet PLOS ONE’s publication criteria as it currently stands. Therefore, we invite you to submit a revised version of the manuscript that addresses the points raised during the review process.

We would appreciate receiving your revised manuscript by Mar 13 2020 11:59PM. To enhance the reproducibility of your results, we recommend that if applicable you deposit your laboratory protocols in protocols.io, where a protocol can be assigned its own identifier (DOI) such that it can be cited independently in the future. For instructions see: http://journals.plos.org/plosone/s/submission-guidelines#loc-laboratory-protocols

We look forward to receiving your revised manuscript.

Kind regards,

Kandasamy Ulaganathan

Academic Editor

PLOS ONE

Journal Requirements:

No

Reviewers' comments:

Reviewer's Responses to Questions

**Comments to the Author**

1. Is the manuscript technically sound, and do the data support the conclusions?

Reviewer #1: Yes

Reviewer #2: Partly

2. Has the statistical analysis been performed appropriately and rigorously? 

Reviewer #1: Yes

Reviewer #2: N/A

3. Have the authors made all data underlying the findings in their manuscript fully available?

Reviewer #1: Yes

Reviewer #2: No

4. Is the manuscript presented in an intelligible fashion and written in standard English?

Reviewer #1: Yes

Reviewer #2: Yes

5. Review Comments to the Author

Reviewer #1: This work is valuable and well written. Authors combined the development of a comprehensive diagnostic tool with field surveys to validate field application and for epidemiological studies. I recommend acceptance with very minor revision.

Line

33: Were congruent with

35: Was P. fuscovaginae expected to be observed on leaves? Since it’s seed/sheath associated?

45: average increase of 4.6% each year

62: Xanthomonas oryzae: X. oryzae pv. oryzae (Xoo) and X. oryzae pv. oryzicola (Xoc)

79: Consider expanding a bit here on the importance of avoiding misidentification of less critical pathogens. Further, since X. oryzae are highly regulated and even considered select agents in the United States, the application of these tools are widely needed.

93: Faso [27] – space needed

95-96: Latin America, however, make the development of an efficient diagnostic tool to detect the disease urgent if it were to gain importance in Africa.

130: selected one pair of

199: Perhaps in the discussion you can propose why the sensitivity levels were higher for B. gladioli and glumae

271: allow a better understanding

Could this assay be converted to Real Time? Likely so, I think authors should discuss this transition and propose as future work for countries and companies with that capability.

Table 3: Formatting could be improved for readability

Reviewer #2: Reviewer Comments:

The manuscript entitled “Design of a new multiplex PCR assay for rice pathogenic bacteria detection and its application to infer disease incidence and detect co-infection in rice fields in Burkina Faso” by Bangratz M. et al., indicate multiplexing technique used for identification of bacterial pathogen in the rice field. The authors have identified unique refence primers for identification of five texa of plant pathogens viz. Pseudomonas, Xanthomonas, Burkholderia, Sphingomonas, and Pantoea species. The authors standardized PCR condition for multiplexing and used the method to identify bacterial infestation in rice field at Burkina Faso. However, there are some severe concerns about the manuscript that need to be addressed for manuscript to meet applicable standards for publication.

Major comments:

1. The authors claim that infestation of Sphingomonas or Pantoea spp. leads to bacterial leaf blight (BLB) like phenotype. However, there are not clear evidences to claim it. Please provide some images of leaves showing such BLB like phenotype. It will be good if authors can provide images of phenotype and indicate in which leaf which bacteria/s were observed. Xanthomonas oryzae pv. oryzae is known to cause BLB symptom (disease lesion in the mid-vein). Did the authors perform leaf-clip inoculation experiment and seen the similar BLB like phenotype with Sphingomonas or Pantoea spp.?

2. The multiplex PCR is the major finding of this manuscript. However, there is not even a single gel image indicating to diagnose this in the field/infected plants. A figure (gel picture) in main text must be added where template genomic DNA is from the infected plant samples (This should be shown for at least KA02 and KA09: where 3 pathogens have been detected).

3. Authors detected only 3 pathogens in the field samples. Is it because of sensitivity of the assay or absence of the pathogen?

Authors should infect plants with the pathogen individually, pool the infected parts, isolate genomic DNA and perform the multiplex PCR.

4. Line 32-33 and 261-262 and Table 3 data: The data in Table 3 indicates the BLB phenotype is majorly because of Sphingomonas spp. As most of the fields showing BLB phenotype are negative for Xanthomonas in molecular diagnostic (Table 3). In the abstract (lines 32-33) and discussion (Lines 261-262) it is reported that “Xanthomonas oryzae incidence levels were in congruence with bacterial leaf streak (BLS) and bacterial leaf blight (BLB) symptom observations in the field” .

5. In Table 3, BLB and BLS column values (such as 17, 95 and 40) are not clear. If 16 plants were observed in the field, what these numbers represent?

6. Reduce size of introduction. Precisely reduce the part where details of all diseases is provided (line 61-96). In the last paragraph of introduction, add little details of findings of this manuscript.

7. Why there is such a huge variation in primer concentration used for different species?

Minor comments

1. Line 230; mention ‘42 strains of five bacterial texa’.

6. PLOS authors have the option to publish the peer review history of their article (what does this mean?). If published, this will include your full peer review and any attached files.

Reviewer #1: Yes: Jillian M. Lang

Reviewer #2: No

---

## [Author Response · Author response to Decision Letter 0]

4 Mar 2020

Dear,

Please find below the reviewer’s comments in black, and our reponses in blue and bold. Line numbers refer to the manuscript with no track changes.

Reviewer #1: 

This work is valuable and well written. Authors combined the development of a comprehensive diagnostic tool with field surveys to validate field application and for epidemiological studies. I recommend acceptance with very minor revision.

Line

L33: Were congruent with

Done.

L35: Was P. fuscovaginae expected to be observed on leaves? Since it’s seed/sheath associated?

We agree with the reviewer that Pseudomonas fuscovaginae induces specific symptoms on sheath and grain, and not on the leaves. However, the bacterium was shown to be epiphytic and endophytic, and could be detected in rice roots, sheath and leaves (Adorada et al 2015 Plant pathology). We consequently added this reference and discussed this point in the discussion section of the manuscript: L251-254.

45: average increase of 4.6% each year

Done.

62: Xanthomonas oryzae: X. oryzae pv. oryzae (Xoo) and X. oryzae pv. oryzicola (Xoc)

Done.

79: Consider expanding a bit here on the importance of avoiding misidentification of less critical pathogens. Further, since X. oryzae are highly regulated and even considered select agents in the United States, the application of these tools are widely needed.

We completed the sentence to clarify our point. See L77-79: “The importance of Pantoea and Sphingomonas species as rice pathogens in West Africa, compared to the well-known devastating Xanthomonas bacteria, remains to be documented and specific molecular detection is critical tool to this purpose.”

93: Faso [27] – space needed

Done.

95-96: Latin America, however, make the development of an efficient diagnostic tool to detect the disease urgent if it were to gain importance in Africa.

Done.

130: selected one pair of

Done.

199: Perhaps in the discussion you can propose why the sensitivity levels were higher for B. gladioli and B. glumae

We don’t have convincing explanation for the higher sensitivity level for B. glumae and B. gladioli compared to other targeted pathogens, so we preferred not to raise this point in the discussion.

271: allow a better understanding

Done.

Could this assay be converted to Real Time? Likely so, I think authors should discuss this transition and propose as future work for countries and companies with that capability.

We agree with the reviewer that many detection methods are now involving real time PCR and that it could have been added to the discussion.

However, the assay presented here could hardly be converted to real time using the same primers. Indeed, our amplicon size range between 263 and 710, while small amplicons (150 or 200bp at maximum) are required for real-time PCR. Consequently, and because we keep with our goal to design detection methods easily applicable in less equipped labs of southern countries (with INERA lab in Burkina Faso as an example), we preferred not to add this information in the discussion.

Table 3: Formatting could be improved for readability

We slightly changed the format and feel the readability is improved.

Reviewer #2: 

The manuscript entitled “Design of a new multiplex PCR assay for rice pathogenic bacteria detection and its application to infer disease incidence and detect co-infection in rice fields in Burkina Faso” by Bangratz M. et al., indicate multiplexing technique used for identification of bacterial pathogen in the rice field. The authors have identified unique reference primers for identification of five taxa of plant pathogens viz. Pseudomonas, Xanthomonas, Burkholderia, Sphingomonas, and Pantoea species. The authors standardized PCR condition for multiplexing and used the method to identify bacterial infestation in rice field at Burkina Faso. However, there are some severe concerns about the manuscript that need to be addressed for manuscript to meet applicable standards for publication.

Major comments:

1. The authors claim that infestation of Sphingomonas or Pantoea spp. leads to bacterial leaf blight (BLB) like phenotype. However, there are not clear evidences to claim it. Please provide some images of leaves showing such BLB like phenotype. It will be good if authors can provide images of phenotype and indicate in which leaf which bacteria/s were observed. Xanthomonas oryzae pv. oryzae is known to cause BLB symptom (disease lesion in the mid-vein). Did the authors perform leaf-clip inoculation experiment and seen the similar BLB like phenotype with Sphingomonas or Pantoea spp.?

The work presented did not aim to show the symptom expression of any of the studied bacterial disease, nor to make a new disease report of BLB caused by Sphingomonas or Pantoea spp in Burkina Faso. Instead, we follow-up previously published work showing that Pantoea and Sphingomonas spp are phytopathogenic for rice in Africa (including Burkina Faso) and that the symptoms are BLB-likes. 

In particular, three articles cited in the manuscript (see the references below, numbers 14, 15 and 18 in the reference list) present some pictures of Pantoea or Sphingomonas infected leaves :

Doni F, Suhaimi NSM, Mohamed Z, Ishak N, Mispan MS. Pantoea: a newly identified causative agent for leaf blight disease in rice. Journal of Plant Diseases and Protection. 2019;126(6):491-4. doi: 10.1007/s41348-019-00244-6.

Kini K, Agnimonhan R, Dossa R, Soglonou B, Gbogbo V, Ouedraogo I, et al. First report of Sphingomonas sp. causing bacterial leaf blight of rice in Benin, Burkina Faso, The Gambia, Ivory Coast, Mali, Nigeria, Tanzania and Togo. New Disease Reports. 2017;35:32. doi: 10.5197/j.2044-0588.2017.035.032.

Kini TK, Agnimonhan R, Afolabi O, Soglonou B, Silue D, Koebnik R. First Report of a New Bacterial Leaf Blight of Rice Caused by Pantoea ananatis and Pantoea stewartii in Togo. Plant Disease. 2017;101(1):241-2. doi: 10.1094/pdis-06-16-0939-pdn. 

In addition to this, below are some citations of articles describing the symptoms associated with Pantoea or Sphingomonas infected leaves :

1) “Lesions appeared first as water-soaked stripes or light brown-to-slightly reddish spots on the upper blades of the leaves, ultimately causing leaf blight” (Lee et al 2010 First Report of leaf blight caused by Pantoea agglomerans on rice in Korea. Plant Disease. 94: 1372–1372).

2) “The disease was thought to be caused by Xanthomonas oryzae pv. oryzae, the rice bacterial blight pathogen. However, physiological and molecular analysis of two strains (ITCC B0050 and ITCC B0055) isolated in 2008 revealed that the causal agent was the bacterium Pantoea ananatis” (Mondal KK, et al 2011. A New leaf blight of rice caused by Pantoea ananatis in India. Plant Disease;95: 1582–1582).

3) “we collected leaf samples from Oryza sativa with clear symptoms of severe leaf blight in experimental fields” (González et al 2014 First report of Pantoea agglomerans causing rice leaf blight in Venezuela. Plant Disease. 99: 552–552)

4) “The biochemical and molecular analysis revealed that the causal agent was not Xanthomonas oryzae pv oryzae, but a new species of bacterium namely Pantoea stewartii subsp. indolegenes.” (Vinodhini et al 2017 Characterization of new bacterial leaf blight of rice caused by Pantoea stewartii subsp. indologenes in Southern Districts of Tamil Nadu. IJEAB. 2: 239027.)

5) “Symptoms included yellow-brown discolourations along one of the two leaf blades, turning brown to dark-brown with age (Fig. 1). Severely affected leaves developed necrotic patches and died” […] “Initial disease symptoms appeared five days after inoculation (DAI), the leaf blade turned yellowish above the inoculation point and this progressed towards the leaf tip (Fig. 3). Blighted leaves, brown to dark-brown necrosis on the entire leaves above and sometimes below the inoculation point, were observed 15-21 DAI on susceptible rice accessions” (Kini K, Agnimonhan R, Dossa R, Soglonou B, Gbogbo V, Ouedraogo I, et al. First report of Sphingomonas sp. causing bacterial leaf blight of rice in Benin, Burkina Faso, The Gambia, Ivory Coast, Mali, Nigeria, Tanzania and Togo. New Dis Reps. 2017;35: 32–32)

Based on this literature review, we argue in the article that Pantoea sp and Sphingomonas sp are phytopathogenic bacteria of rice and that associated symptoms may be confounded with those caused by X. oryzae pv oryzae (Xoo), and so that the symptoms can be referred to as bacterial leaf blight (BLB)-like.

We modified the introduction part to make the point more clear (L70-79) and we added two references (Lee et al 2010 Plant Disease, and González et al 2014 Plant Disease) to complement the literature review.

2. The multiplex PCR is the major finding of this manuscript. However, there is not even a single gel image indicating to diagnose this in the field/infected plants. A figure (gel picture) in main text must be added where template genomic DNA is from the infected plant samples (This should be shown for at least KA02 and KA09: where 3 pathogens have been detected).

We agree it would improve the manuscript to include a gel image showing the PCR multiplex applied on field samples from Burkina Faso. However, since we already have two figures of gel images in main text, we preferred to put the additional figure as supplementary material (S2Fig).

In this new figure, we show as exemples six field samples representing each obtained cases: 1) no bacteria, 2) Sphingomonas spp, Xanthomonas oryzae, and Pantoea spp. 3) Sphingomonas spp. only, 4) Sphingomonas spp and Xanthomonas oryzae, 5) Sphingomonas spp and Pantoea spp and 6) Xanthomonas oryzae only. Reference to this figure appears L215 in the main text.

3. Authors detected only 3 pathogens in the field samples. Is it because of sensitivity of the assay or absence of the pathogen?

Authors should infect plants with the pathogen individually, pool the infected parts, isolate genomic DNA and perform the multiplex PCR.

We are confident in the sensitivity of our method (see Figure 2) and consequently, we think that Pseudomonas fuscovaginae and Burkholderia glumae are not present in the 256 analysed leaf samples. We clarified this point in the discussion (L250-251).

As pointed out by reviewer 1, Pseudomonas fuscovaginae leads to specific symptoms on sheath and grain, and not on the leaves. However, the bacterium was shown to be epiphytic and endophytic, and could be detected in rice roots, sheath and leaves (Adorada et al 2015 Plant pathology). Actually, Pseudomonas fuscovaginae is known to be present in 31 countries (BABI 2007 cited in Bigirimana et al 2015 Frontiers in Plant Science), but not any of these 31 countries are in West Africa. Consequently, it was actually quite unlikely to find P. fuscovaginae in our field samples from Burkina Faso.

Similarly, Burkholderia glumae and B. gladioli have been mostly isolated from infected panicle. Only two studies reported these bacteria in Africa and none of them includes molecular data. So it would also have been a surprise to find Burkholderia glumae and B. gladioli in our field samples from Burkina Faso.

We add a paragraph discussing these points in the manuscript: L249-258.

We performed preliminary experimental infections and analysis of infected leaf samples for Xanthomonas oryzae, the bacteria we have been mostly working with in our lab and it worked with no problem. We did not follow-up this methodology but instead compared the DNA obtained from bacterial cultures to DNA from bacterial culture with addition of plant DNA and sensitivity did not change. This is stated in the manuscript L199-200: “Addition of 250 ng plant DNA to bacterial DNA did not change the amplification results.”

4. Line 32-33 and 261-262 and Table 3 data: The data in Table 3 indicates the BLB phenotype is majorly because of Sphingomonas spp. As most of the fields showing BLB phenotype are negative for Xanthomonas in molecular diagnostic (Table 3). In the abstract (lines 32-33) and discussion (Lines 261-262) it is reported that “Xanthomonas oryzae incidence levels were in congruence with bacterial leaf streak (BLS) and bacterial leaf blight (BLB) symptom observations in the field”.

We do not agree with the following assertion “The data in Table 3 indicates the BLB phenotype is majorly because of Sphingomonas spp. As most of the fields showing BLB phenotype are negative for Xanthomonas in molecular diagnostic (Table 3)”. Indeed, among the 16 studied field, 11 fields had BLB symptom observations but in four of them the incidence estimate is very low (1%), and if considering the seven fields with more than 2% BLB incidence estimate, four of them (more than half) had some plants positive for Xanthomonas oryzae in multiplex PCR. Three of these four fields had no BLS symptoms and consequently Xo molecular detection is likely attributed to Xanthomonas oryzae pv oryzae. 

We tried to make this point on the relationship between molecular detection and symptoms more clear in the manuscript by adding a new additional figure (S3 Figure). Also, the text has been updated to be clearer on the point of relationship between symptoms and molecular detection results: see in the results section L228-232: “Apart from these two fields, the highest Xo incidence levels (3/16 = 19%) were found in two fields (BZ06 and BZ12) also has relatively high symptom-based BLB estimates (ca 10%; Table 3 and S3 Fig). We found no clear relationship between BLB symptom-based incidence estimates and either Sphingomonas or Pantoea molecular incidence estimates (Table 3).” We added a reference to the new additional figure in the discussion part (see the paragraph on this topic L274-282).

Finally, we agree with the reviewer that molecular detection data and symptom observations were not perfectly congruent and we consequently slightly changed the sentence in the abstract (L33), we changed “were congruent” by “were mostly congruent”. 

5. In Table 3, BLB and BLS column values (such as 17, 95 and 40) are not clear. If 16 plants were observed in the field, what these numbers represent?

The incidence estimates are based on symptom observation in four (5meters*5meters) cells of the diagonal of the grid. This is mentioned in the material and methods section L171-173: “Symptom-based incidence was estimated in each field for BLB and BLS by carefully observing plants in the four cells along the diagonal of the 4x4 grid (obtained average incidence resulted from the average of recorded incidence levels over the four cells).”

We complemented the legend of Table 3 to make it clearer, see L329-332: “Table 3: Obtained results in the 16 fields surveyed in Southern Burkina Faso: pathogen incidences derived from the use of the developed molecular diagnostic tool on 16 sampled plants per field, and disease incidence estimated from symptom observations in four cells of the field’s diagonal”.

6. Reduce size of introduction. Precisely reduce the part where details of all diseases is provided (line 61-96). In the last paragraph of introduction, add little details of findings of this manuscript.

We agree with the reviewer that the part of the introduction describing the targeted bacterial diseases was quite long and we followed the recommendation to reduce it. In particular, various sentences were shortened, and one was removed (“The genus Burkholderia comprises several rice pathogenic bacteria, while the sister genus Paraburkholderia includes phytobeneficial species”). See L61-93.

Also, as recommended, we add two sentences stating the major findings of the manuscript at the end of the introduction, see L107-112.

7. Why there is such a huge variation in primer concentration used for different species?

We agree with the reviewer that there is important variation in primer concentration. The primer concentration presented in the article are the result of a gradual empirical adjustment of relative concentrations to obtain comparable sensitivity for each taxa.

Minor comments

1. Line 230; mention ‘42 strains of five bacterial texa’.

Done.

---

## [Decision Letter · Decision Letter 1]

8 Apr 2020

Design of a new multiplex PCR assay for rice pathogenic bacteria detection and its application to infer disease incidence and detect co-infection in rice fields in Burkina Faso

PONE-D-19-32664R1

Dear Dr. Tollenaere,

We are pleased to inform you that your manuscript has been judged scientifically suitable for publication and will be formally accepted for publication once it complies with all outstanding technical requirements.

With kind regards,

Kandasamy Ulaganathan

Academic Editor

PLOS ONE

Additional Editor Comments (optional):

Reviewers' comments:

Reviewer's Responses to Questions

**Comments to the Author**

1. If the authors have adequately addressed your comments raised in a previous round of review and you feel that this manuscript is now acceptable for publication, you may indicate that here to bypass the “Comments to the Author” section, enter your conflict of interest statement in the “Confidential to Editor” section, and submit your "Accept" recommendation.

Reviewer #1: All comments have been addressed

2. Is the manuscript technically sound, and do the data support the conclusions?

Reviewer #1: Yes

3. Has the statistical analysis been performed appropriately and rigorously? 

Reviewer #1: Yes

4. Have the authors made all data underlying the findings in their manuscript fully available?

Reviewer #1: Yes

5. Is the manuscript presented in an intelligible fashion and written in standard English?

Reviewer #1: Yes

6. Review Comments to the Author

Reviewer #1: (No Response)

7. PLOS authors have the option to publish the peer review history of their article (what does this mean?). If published, this will include your full peer review and any attached files.

Reviewer #1: Yes: Jillian M. Lang

---

## [Editor Report · Acceptance letter]

15 Apr 2020

PONE-D-19-32664R1 

Design of a new multiplex PCR assay for rice pathogenic bacteria detection and its application to infer disease incidence and detect co-infection in rice fields in Burkina Faso 

Dear Dr. Tollenaere:

I am pleased to inform you that your manuscript has been deemed suitable for publication in PLOS ONE. Congratulations! Your manuscript is now with our production department. 

With kind regards,

on behalf of

Dr. Kandasamy Ulaganathan 

Academic Editor

PLOS ONE